# Resting Energy Expenditure in Older Inpatients: A Comparison of Prediction Equations and Measurements

**DOI:** 10.3390/nu14245210

**Published:** 2022-12-07

**Authors:** Fumiya Kawase, Yoshiyuki Masaki, Hiroko Ozawa, Manami Imanaka, Aoi Sugiyama, Hironari Wada, Ryokichi Goto, Shinya Kobayashi, Takayoshi Tsukahara

**Affiliations:** 1Department of Nutrition, Asuke Hospital Aichi Prefectural Welfare Federation of Agricultural Cooperatives, 20 Nakata, Yagami-cho, Toyota 444-2351, Aichi, Japan; 2Graduate School of Nutritional Sciences, Nagoya University of Arts and Sciences, 57, Iwasaki-cho, Nisshin 470-0196, Aichi, Japan; 3Department of Internal Medicine, Asuke Hospital Aichi Prefectural Welfare Federation of Agricultural Cooperatives, 20 Nakata, Yagami-cho, Toyota 444-2351, Aichi, Japan; 4Department of Community-Based Medical Education, Graduate School of Medical Sciences, Nagoya City University, Mizuho-ku, Nagoya 467-8601, Aichi, Japan; 5Department of Nursing, Asuke Hospital Aichi Prefectural Welfare Federation of Agricultural Cooperatives, 20 Nakata, Yagami-cho, Toyota 444-2351, Aichi, Japan; 6Department of Rehabilitation Therapy, Asuke Hospital Aichi Prefectural Welfare Federation of Agricultural Cooperatives, 20 Nakata, Yagami-cho, Toyota 444-2351, Aichi, Japan

**Keywords:** resting energy expenditure, prediction accuracy, older patients, indirect calorimetry

## Abstract

Determining energy requirements are an important component of nutritional support for patients with malnutrition; however, the validity of prediction equations for resting energy expenditure (REE) is disputed in older hospitalized patients. We aimed to assess the validity of these equations in older hospitalized patients in Japan. This was a single-center, cross-sectional study of 100 patients aged ≥70 years, hospitalized between January 2020 and December 2021. REE was measured using an indirect calorimeter and was compared to the predicted values calculated from five REE prediction equations. The mean (95% confidence interval) measured REE was 968.1 (931.0, 1005.3) kcal/day, and the mean predicted REE was higher for the FAO/WHO/UNU (1014.3 [987.1, 1041.6] kcal/day, *p* = 0.164) and Schofield (1066.0 [1045.8, 1086.2] kcal/day, *p* < 0.001) equations and lower for the Harris-Benedict (898.6 [873.1, 924.1] kcal/day, *p* = 0.011), Ganpule (830.1 [790.3, 869.9] kcal/day, *p* < 0.001), and body weight (kg) × 20 (857.7 [821.9, 893.5] kcal/day, *p* < 0.001) equations. In the age group analysis, none of the predicted values were within a 10% error for more than 80% of patients aged 70–89 years and ≥90 years. The five REE prediction equations did not provide accurate estimates. Validated REE prediction equations need to be developed for older hospitalized patients.

## 1. Introduction

Malnutrition is recognized as a global issue with a prevalence of 0.8–24.6% in the increasing global aging population [1]. Malnutrition is associated with adverse clinical outcomes including prolonged hospital stays [2], increased in-hospital mortality [3], and increased in-hospital falls [4]. Thus, interventions for malnutrition have the potential to improve patient care and quality of life and reduce healthcare costs [5]. The EFFORT Trials [6,7], also known as the Effect for Protein Dosing in Critically Ill Patients study, examined the effects of personalized nutrition management on hospitalized patients, and showed reduced mortality (Hazard Ratio 0.65) after providing nutrition support to 2088 patients. A key component of the positive impacts of the EFFORT trials was the assessment of energy requirements using indirect calorimetry or the Harris-Benedict equation. Energy intake is one of the most important factors to consider in providing nutritional support, and inappropriate energy intake is associated with adverse outcomes. This has been reported in critically ill patients [8], as well as older patients with diabetes [9]. Therefore, a more accurate calculation of daily energy requirements are required for all patients.

Resting energy expenditure (REE) comprises the largest component of total energy expenditure and is, therefore, the most important determinant of its value [10]. In clinical practice, energy requirements are generally calculated by multiplying the estimated REE by physical activity and disease coefficients [11] and are rarely calculated from measured values. REE can be measured using indirect calorimetry, which is a noninvasive method and is often considered the gold standard for deriving REEs. However, indirect calorimetry is not routinely available in many healthcare or primary care facilities because of the high cost of equipment and technical complexity [12]. Therefore, several prediction equations have been developed to predict REE. The Harris-Benedict equation [13] is a commonly used REE prediction equation in clinical practice [14]; however, there are conflicting reports regarding the accuracy of equations in predicting REE in older hospitalized patients [15,16]. Thus, there is no consensus on the utility of the Harris-Benedict equation for older adults. In general, most REE prediction equations have been developed for healthy adult populations, and their predictive accuracy among older adults has been questioned [10]. A systematic review of healthy older adults showed that several REE prediction equations, including the Harris-Benedict equation, predicted a larger REE value than the measured REE values, suggesting that accurate estimation using prediction equations is difficult to obtain for older adults [17]. Therefore, in nutritional management guidelines for older adults, it is not recommended to use REE prediction equations to determine energy requirements [11,18]. This guideline recommends a Total energy expenditure of approximately 30 kcal/kg, since the REE of older adults is approximately 20 kcal/kg.

The patterns in prediction accuracy and error of REE prediction equations used for older Japanese hospitalized patients are unknown. The accuracy of several predictive equations is lower in specific populations, such as older hospitalized patients, and has not been validated for use in Japanese older patients. Therefore, in this study, we aimed to investigate the predictive accuracy of REE prediction equations in older hospitalized patients by comparing the calculated predicted REE values with the measured REE values obtained using indirect calorimetry.

## 2. Materials and Methods

### 2.1. Study Design and Participants

A single-center, cross-sectional study was conducted in a 200-bed general hospital on older patients admitted to the internal medicine ward between January 2020 and December 2021. This study was approved by the Ethical Review Committee for Clinical Trials and Clinical Research, Aichi Prefectural Welfare Federation of Agricultural Cooperatives, Asuke Hospital (registration number: K19-009) and conducted in accordance with the Declaration of Helsinki and the Japanese Ethical Guidelines for Medical Research Involving Human Subjects (Ministry of Education, Culture, Sports, Science and Technology and Ministry of Health, Labour and Welfare). This study has been registered in the UMIN Clinical Trials Registration System (UMIN study ID: UMIN000040499).

Eligible patients included those who: (1) were aged ≥70 years, (2) were hospitalized at the Department of Internal Medicine, and (3) consented to participate in all procedures related to the study. The exclusion criteria were: those (1) receiving oxygen therapy or being mechanically ventilated, (2) undergoing enteral nutrition management, (3) undergoing dialysis management, (4) with the presence of respiratory leaks, and (5) with the presence of poor outcomes whose hospital stay was estimated to be less than one week by the attending physician. Patients who could not complete indirect calorimetry and the body composition tests using the multifrequency bioelectric impedance analysis (BIA) method and who developed acute illness during hospitalization and were transferred to another hospital were also excluded from the analysis. Indirect calorimetry, body composition test, and diagnosis of malnutrition were performed within approximately one week of obtaining written informed consent from the patients. 

### 2.2. Indirect Calorimetry

REE was measured using an indirect calorimetry device (AE-300S, Minato Medical Science, Osaka, Japan). Indirect calorimetry was performed in a hospital room after an overnight (12 h) fast. Calibrations of flow rate and standard gas were performed before every measurement, and the room temperature was controlled at 23–25 °C. Patients rested for 30 min before undergoing measurements for 15 min between 7:00 and 8:00 a.m. A pump was used to suction through the hood at a constant rate. Using the Weir equation (Equation (1)) [19], the 24 h REE was calculated from the measured oxygen consumption (VO_2_) and carbon dioxide production (VCO_2_).
(1)$$\mathrm{REE}=\left(\;3.94\times {\;\dot{V}O}_{2}+1.11\times {\;\dot{V}\mathrm{CO}}_{2}\;\right)\times 1440$$

### 2.3. Measurements and Body Composition Analysis

Patients were evaluated using the Charlson Comorbidity Index [20] and Mini Nutritional Assessment-Short Form (MNA-SF) score [21] on the same day as indirect calorimetry. Body composition was evaluated by the BIA method using a body composition analyzer (Inbody S10, Inbody Co, Seoul, Republic of Korea) and calf circumference. BIA measurements were obtained from patients who had fasted overnight (12 h) and were in a supine position at rest for at least 15 min after urination. After the BIA measurements were obtained, the skeletal muscle index (SMI) was calculated. Calf circumference was measured at the thickest part of the leg opposite to the dominant leg with the right leg bent at 90° in the supine position.

### 2.4. Global Leadership Initiative on Malnutrition (GLIM) Criteria

The Global Leadership Initiative on Malnutrition (GLIM) criteria were used to diagnose malnutrition [22]. All patients were screened for malnutrition based on the MNA-SF score and were asked about weight change in the past 3–6 months. Body mass index (BMI) was calculated, and muscle mass was assessed by the SMI calculated using the BIA method. Dietary intake or decreased digestion and absorption were assessed based on information from just before and several days after admission. To assess malnutrition severity, a low BMI was assessed using the cutoff value of 17.8 kg/m^2^ for older Japanese adults [3], and severe muscle mass loss was defined as a decrease of 10% or more below the SMI value of <7.0 kg/m^2^ in men and <5.7 kg/m^2^ in women specified by the Asian Working Group for Sarcopenia 2019 consensus update [23].

### 2.5. Predictive Equations for REE

The five prediction equations used in this analysis were: (1) Harris-Benedict [13], (2) FAO/WHO/UNU [24], (3) Ganpule [25], (4) Schofield [26], and (5) body weight (kg) × 20. The first four are REE prediction equations discussed in the Dietary Reference Intakes for Japanese (2020) [27], and the fifth one is given in the European Society for Clinical Nutrition and Metabolism guidelines on clinical nutrition and hydration in geriatrics [11]. Characteristics of each equation are presented in Table 1.

### 2.6. Statistical Analysis

Previous study reported a difference of 150 ± 103 kcal/day between the Harris-benedict equation and the measured REE [28], so this value was used to calculate the sample size in advance. Therefore, using power = 0.8 and α error = 0.01 by Bonferroni correction, more than 60 patients were needed.

Statistical analysis was performed in three groups: overall, aged 70-89, and aged 90 and older.

Parametric continuous variables are expressed as means ± SD, and nonparametric continuous variables are expressed as medians. One-way analysis of variance was used to compare measured REE and predicted REE, and multiple comparison tests with measured REE as a control were conducted as post hoc tests. R 4.1.0 (R Foundation for Statistical Computing, Vienna, Austria) was used for all analyses [29].

## 3. Results

In this study, 100 of the 127 patients who participated completed the measurements. Table 2 shows the characteristics of the 100 patients. For all patients, the mean age was 88.1 ± 6.8 years, and 34% were male. The mean BMI was 20.1 ± 3.5 kg/m^2^, and the mean SMI was 4.67 ± 1.47 kg/m^2^. According to the GLIM criteria, the proportions of severe and moderate malnutrition were 72% and 28%, respectively.

### 3.1. Accuracy of the REE Prediction Equations

The results for the measured REE and calculated REE values from each of the prediction equations are shown in Table 3 and Figure 1A. For the entire study group, the mean (95% confidence interval) value of the measured REE was 968.1 (931.0, 1005.3) kcal/day. When compared with the measured REE, the Harris-Benedict, Ganpule, and body weight (kg) × 20 equations showed lower REE values (898.6 [873.1, 924.1] kcal/day, *p* = 0.011; 830.1 [790.3, 869.9] kcal/day, *p* < 0.001; 857.7 [821.9, 893.5] kcal/day, *p* < 0.001; respectively). The FAO/WHO/UNU and Schofield equations led to higher REE values (1014.3 [987.1, 1041.6] kcal/day, *p* = 0.164; 1066.0 [1045.8, 1086.2] kcal/day, *p* < 0.001; respectively). However, the trends for the FAO/WHO/UNU equation were not statistically significant. In the 70–89 years age group, the mean measured REE was 1041.8 (987.5, 1096.1) kcal/day; when compared to the measured REE, the Harris-Benedict, Ganpule, and body weight (kg) × 20 equations calculated lower REE values (964.9 [929.7, 1000.1] kcal/day, *p* = 0.068; 924.2 [870.5, 977.8] kcal/day, *p* < 0.001; 929.0 [880.0, 978.1] kcal/day, *p* = 0.002; respectively), and the FAO/WHO/UNU and Schofield equations calculated higher values (1074.5 [1034.9, 1114.0] kcal/day, *p* = 0.761; 1105.5 [1077.5, 1133.6] kcal/day, *p* = 0.174; respectively). However, the trends for the Harris-Benedict, FAO/WHO/UNU, and Schofield equations were not statistically significant. In the 90+ years age group, the measured REE was 891.4 (850.3, 932.6) kcal/day. When compared to the measured REE, the Harris-Benedict, Ganpule, and body weight (kg) × 20 equations calculated lower values (829.6 [804.3, 854.9] kcal/day, *p* = 0.071; 732.3 [687.1, 777.5] kcal/day, *p* < 0.001; 783.5 [739.7, 827.2] kcal/day, *p* < 0.001; respectively), and the FAO/WHO/UNU and Schofield equations calculated higher values (951.8 [923.1, 980.4] kcal/day, *p* = 0.081; 1024.8 [1000.3, 1049.3] kcal/day, *p* < 0.001; respectively). However, the trends for the Harris-Benedict and FAO/WHO/UNU equations were not statistically significant. 

### 3.2. Individual-Level Bias of the REE Prediction Equation

During examination of the error of the predicted REE compared to the actual measured REE, we found that none of the prediction equations yielded a predicted REE within a 10% error of the measured REE for more than 80% of individuals in either age group (Figure 1B). For the entire study group, the proportion of individuals within a 10% error was 51% for the Harris-Benedict equation, 59% for the FAO/WHO/UNU equation, 30% for the Ganpule equation, 38% for the Schofield equation, and 35% for the body weight (kg) × 20 equation. For patients aged 70–89 years, the proportion of individuals within a 10% error was 53% for the Harris-Benedict equation, 65% for the FAO/WHO/UNU equation, 37% for the Ganpule equation, 45% for the Schofield equation, and 35% for the body weight (kg) × 20 equation; for patients aged 90 years or older, the proportion of individuals within a 10% error was 49% for the Harris-Benedict equation, 53% for the FAO/WHO/UNU equation, 22% for the Ganpule equation, 31% for the Schofield equation, and 35% for the body weight (kg) × 20 equation. From these findings, the error in predicted REE values was higher for patients aged 90+ years. 

## 4. Discussion

In this study, REE was measured by indirect calorimetry, which is considered the gold-standard method of measuring REE, in 100 Japanese patients aged 70–101 years, and body composition was measured using the validated BIA method. We found that the currently available REE prediction equations had clinically relevant errors relative to the measured REE for older Japanese people. Additionally, the error was larger for patients aged over 90 years, which is an important finding to consider for the energy requirements of hospitalized older patients.

The Harris-Benedict equation is commonly used for REE prediction in clinical practice in Japan. However, our study revealed that this equation predicted a value that was, on average, 69.5 kcal/day less than the measured REE. Furthermore, compared to the measured REE, the FAO/WHO/UNU equation and Schofield equation overestimated by 46.2 kcal/day and 97.9 kcal/day, respectively, while the Ganpule equation and body weight × 20 equation underestimated by 138.0 kcal/day and 110.4 kcal/day, respectively. A previous study, that measured REE by indirect calorimetry in 68 hospitalized patients aged 74.4 ± 9.3 years, reported that the predicted REE computed by the Harris-Benedict equation was 26.7 kcal/day lower than the measured REE, displaying a smaller error than our findings [30]. Furthermore, Siervo et al. study, the FAO/WHO/UNU equation was 105 kcal/day higher than the measured REE, while the Schofield equation was 7.2 kcal/day lower, and the accuracy of the Schofield equation seems to be higher. In our findings, both prediction equations were overestimated, and thus, our study showed a different trend from previous studies. Similarly, Neelemaat et al. assessed indirect calorimetry on 194 hospitalized older patients aged 74.3 ± 9.1 years and found that the predicted REE calculated by the Harris-Benedict equation was 141 kcal/day lower than the measured REE, displaying a larger error than our findings. Furthermore, in the Neelemaat et al. study, the FAO/WHO/UNU equation and the Schofield equation showed smaller values than the measured REE. The authors also concluded that none of the analyzed equations predicted REE with sufficient accuracy in older hospitalized patients in their study [15]. Consistent with these reports, the predicted REE calculated by the Harris-Benedict equation in our study was lower than the actual value measured using indirect calorimetry in older hospitalized patients, although the magnitude of error varied. In contrast, Miyake et al. reported that, in healthy Japanese women aged ≥70 years, the predicted REE calculated by the Harris-Benedict equation was observed to be 86 ± 83 kcal greater than the actual value measured by indirect calorimetry [28]. Their study analyzed a healthy population with a maximum age of 79 years, which is a different population from that in our study and may explain the different values. The authors also concluded that the predicted REE using the Ganpule equation had the highest prediction accuracy compared to the Harris-Benedict, Schofield, and FAO/WHO/UNU equations [28]. This study examined the prediction equations presented in the Dietary Reference Intakes for Japanese (Males; body weight × 21.5, Females; body weight × 20.7), which are similar to the body weight × 20 prediction equation. The results reported values approximately 32 to 80 kcal/day greater than the measured REE. Due to the slightly larger coefficients and the different patient settings, the trends were different from our findings. however, Gaillard et al. observed that the Harris-Benedict equation had the highest estimation accuracy for older patients [31]. In our study, the Harris-Benedict and FAO/WHO/UNU equations showed relatively high estimation accuracy among the prediction equations analyzed, however, the observed errors were too large for the equations to be used in clinical practice. A recent review reported that 210 different REE prediction equations, some of which were developed specifically for older adults, had low agreement as assessed by the intraclass correlation coefficient and large heterogeneity in REE prediction value [32]. Therefore, it is desirable to find suitable prediction equations for various factors in addition to age, such as sex, physique, activity, and morbidity. In a meta-analysis of healthy older adults, the Harris-Benedict equation showed the highest accuracy at the individual level, with 69% of individuals with predicted REE values within a 10% error [17]. Therefore, the feasibility of the Harris-Benedict, FAO/WHO/UNU, Ganpule, Schofield, and body weight × 20 as predictive equations for older hospitalized patients seems to be limited, as there are conflicting reports in the literature.

Additionally, previous reports studying the estimation accuracy of REE prediction equations for cancer patients reported that none of the equations had sufficient accuracy in predicting actual REE, suggesting that accurate individual-level REE prediction is also difficult for patients with disease [33,34]. REE prediction equations have recently been developed for chronic kidney disease patients specifically [35], and recent studies suggest that REE prediction equations may need to be developed for each disease. Our findings have not been evaluated in the context of diseases; therefore, further studies are needed to clarify this relationship.

Adequate energy intake (approximately ≥ 30 kcal/kg/day) has been reported to reduce post-discharge mortality [36] and increase rehabilitation efficiency in patients with dysphagia [37]. In contrast, both overfeeding and underfeeding have been associated with adverse clinical outcomes in critically ill patients [8]. Similarly, a U-shaped relationship between energy intake and mortality was observed in older Japanese patients with diabetes [9]. As accurate estimates are needed to provide good clinical outcomes for hospitalized older patients, unvalidated predictive equations should be avoided.

In a study conducted by Schuetz et al., it was found that individualized nutrition management prevented adverse clinical outcomes in hospitalized patients, and a key factor in obtaining this result was the determination of individualized energy requirements [6]. In older adults, appetite decreases with age, and malnutrition develops rapidly [38]. Individualized nutritional management is a powerful strategy for preventing malnutrition in older patients; however, the findings of our study showed that there was a large error in the predicted REE values calculated from prediction equations. Therefore, to improve clinical outcomes in older hospitalized patients, prediction equations for REE should not be used in a generalized manner, and development of REE prediction equations specific to the disease and patient characteristics is necessary. 

This study had a few limitations. First, this was a single-center cross-sectional study conducted in Japan, which limits the generalization of results due to possible selection bias of participants. Second, since the study included individuals with diverse diseases and backgrounds that were not controlled for, it is possible that our results were confounded by such factors. The impact of disease on REE is unknown even for diseases that have a significant impact on metabolism; therefore, the relationship between disease and REE should be further investigated. Third, in this study, we did not consider the impact of BMI on the accuracy of the REE prediction equation because we focused our analysis on each age group. In most REE prediction equations, BMI components such as height and weight were used as variables in the prediction equation, therefore, future studies are needed to determine the accuracy of the REE prediction equation for each BMI group.

## 5. Conclusions

In this study, in older patients aged 70–101 years, the large error between the actual REE measured by indirect calorimetry and the predicted REE calculated by the five prediction equations indicates that REE prediction equations may not be valid for clinical use. This error was larger in patients over 90 years of age. The results of this study showed that none of the five equations compared had predictive equations with high estimation accuracy in older Japanese hospitalized patients. There is a need to develop REE prediction equations with greater accuracy, especially for the accurate prediction of REE in older patients.

## Figures and Tables

**Figure 1 nutrients-14-05210-f001:**
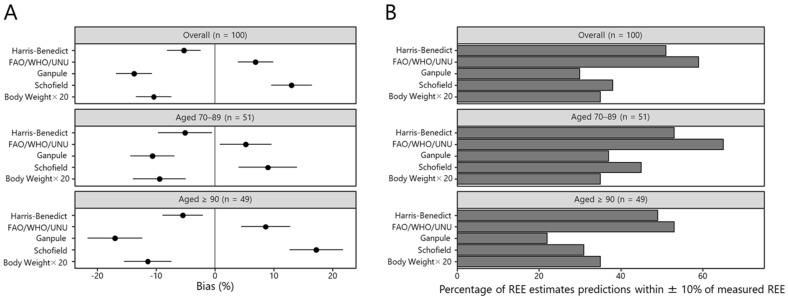
Accuracy of resting energy expenditure (REE) prediction equations by age group. (**A**) Percent bias of resting energy expenditure (REE) prediction equations compared to the measured REE. The mean relative bias (%) was calculated as follows: ((ΔREE_Predicted_)/REE_Measured_) × 100. Error bars represent the 95% confidence interval; (**B**) Percentage of REE predictions with a mean relative bias (%) within 10% of the measured REE for each prediction equation.

**Table 1 nutrients-14-05210-t001:** Characteristics of equations for the prediction of resting energy expenditure (REE).

Equations	Sex	Formula
Harris-Benedict	Male	66.4730 + 13.7516 × W (kg) + 5.0033 × H (cm) − 6.7550 × A
	Female	655.0955 + 9.5634 × W (kg) + 1.8496 × H (cm) − 4.6756 × A
FAO/WHO/UNU	Male	(36.8 × W (kg) + 4719.5 × (H (cm)/100) − 4481)/4.186
	Female	(38.5 ×W (kg) + 2665.2 × (H (cm)/100) − 1264)/4.186
Ganpule	Male	(0.0481 × W (kg) + 0.0234 × H (cm) − 0.0138 × A − 0.4235) × 1000/4.186
	Female	(0.0481 × W (kg) + 0.0234 × H (cm) − 0.0138 × A − 0.9708) × 1000/4.186
Schofield	Male	(0.049 ×W (kg) + 2.459) × 1000/4.186
	Female	(0.038 ×W (kg) + 2.755) × 1000/4.186
Body weight × 20		W (kg) × 20

Abbreviations: W weight, H Height, A Age.

**Table 2 nutrients-14-05210-t002:** Clinical characteristics of the studied patients.

Characteristics	Overall (n = 100)	Aged 70–89 (n = 51)	Aged ≥ 90 (n = 49)
Age, years	88.1 ± 6.8	82.9 ± 5.1	93.5 ± 3.0
Sex, male *n* (%)	34 (34%)	23 (45%)	11 (22%)
Height, cm	146.0 ± 10.7	150.0 ± 11.6	142.0 ± 7.9
Weight, kg	42.9 ± 9.1	46.5 ± 8.9	39.2 ± 7.8
BMI, kg/m^2^	20.1 ± 3.5	20.8 ± 3.6	19.4 ± 3.3
SMI, kg/m^2^	4.67 ± 1.47	4.99 ± 1.42	4.34 ± 1.46
CC, cm	27.3 ± 3.9	28.4 ± 4.1	26.2 ± 3.3
MNA-SF, score	7 (6–10)	8 (6–10)	7 (6–9)
CCI, score	2 (1–3)	2 (1–3)	2 (1–2)
GLIM criteria			
Malnutrition, *n* (%)	76 (76%)	37 (73%)	39 (80%)
Moderate malnutrition, *n* (%)	21 (28%)	11 (30%)	10 (26%)
Severe malnutrition, *n* (%)	55 (72%)	26 (70%)	29 (74%)

Data are expressed as numbers of participants (%), means ± standard deviation, or medians (interquartile range). Abbreviations: BMI Body mass index, SMI Skeletal muscle index, CC Calf circumference, MNA-SF Mini Nutritional Assessment Short Form, CCI Charlson comorbidity index, GLIM Global Leadership Initiative on Malnutrition.

**Table 3 nutrients-14-05210-t003:** Evaluation of resting energy expenditure (REE) predictive prediction equations.

REE Predictive Prediction Equation	Overall (n = 100)	Aged 70–89 (n = 51)	Aged ≥ 90 (n = 49)
	Mean (95% CI)	Mean (95% CI)	Mean (95% CI)
Measured REE, kcal/day	968.1 (931.0, 1005.3)	1041.8 (987.5, 1096.1)	891.4 (850.3, 932.6)
Harris-Benedict, kcal/day	898.6 (873.1, 924.1) *	964.9 (929.7, 1000.1)	829.6 (804.3, 854.9)
FAO/WHO/UNU, kcal/day	1014.3 (987.1, 1041.6)	1074.5 (1034.9, 1114.0)	951.8 (923.1, 980.4)
Ganpule, kcal/day	830.1 (790.3, 869.9) ***	924.2 (870.5, 977.8) **	732.3 (687.1, 777.5) ***
Schofield, kcal/day	1066.0 (1045.8, 1086.2) ***	1105.5 (1077.5, 1133.6)	1024.8 (1000.3, 1049.3) ***
Body weight × 20, kcal/day	857.7 (821.9, 893.5) ***	929.0 (880.0, 978.1) **	783.5 (739.7, 827.2) ***

*** *p* < 0.001, ** *p* < 0.01, * *p* < 0.05, Dunnett tests were performed with reference to the measured REE. Abbreviations: CI confidence interval.

## Data Availability

Data sharing is not applicable to this article.

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
