# Peer review of "Resting Energy Expenditure in Older Inpatients: A Comparison of Prediction Equations and Measurements"

_nutrients, 2022, doi:10.3390/nu14245210_

Round 1

Reviewer 1 Report

In the present MS “Resting Energy Expenditure in Older Inpatients: A Comparison 2 of Prediction Equations and Measurements”, Kawase and et al. presented the information about the difference of REE calculated by different equations in older inpatients in JP. The results show the present five REE prediction equations can not provide accurate REE for older inpatients. Overall, I think that the manuscript is straight forward, and easy to read. However, there are some weak points.

1. Since this study is a cross-sectional study, please use STROBE Checklist as a supporting information. 2. All formula is based on BMI, have authors ever divide groups by BMI and calculate again?

3. Although authors compared the error of five equations, however, only Harris-Benedict equation was discussed. Please discuss more why there are different results calculated by different prediction equations

Author Response

We thank the reviewer for the helpful comments, which assisted us in improving the manuscript. We have made additional changes in the revised manuscript based on the comments. Our changes are indicated with red font in the revised manuscript.

Comment #1: Since this study is a cross-sectional study, please use STROBE Checklist as a supporting information.

Response: Thank you for your valuable comments. We have rechecked and provided a STROBE checklist. As a result, a description of sample size calculations and classification of groupings has been added to the statistical analysis section.

2.6. Statistical analysis

              Previous study reported a difference of 150 ± 103 kcal/day between the Har-ris-benedict equation and the measured REE [28], so this value was used to calculate the sample size in advance. Therefore, using power = 0.8 and α error = 0.01 by Bonferroni correction, more than 60 patients were needed.

Statistical analysis was performed in three groups: overall, aged 70-89, and aged 90 and older. (Page 4, Lines 148–153)

Parametric continuous variables are expressed as means ± SD, and nonparametric continuous variables are expressed as medians. One-way analysis of variance was used to compare measured REE and predicted REE, and multiple comparison tests with measured REE as a control were conducted as post hoc tests. R 4.1.0 (R Foundation for Statistical Computing, Vienna, Austria) was used for all analyses [28]. (Page4, Lines 154–158)

Comment #2: All formula is based on BMI, have authors ever divide groups by BMI and calculate again?

Response: We agree with your comment that all the prediction equations used in our analysis used height or weight, which is a component of BMI. Therefore, we consider the predictive validity of each BMI group to be as important as age. However, since this study focused on predictive validity by age group, the following was added to the Limitation.

Third, in this study, we did not consider the impact of BMI on the accuracy of the REE prediction equation because we focused our analysis on each age group. In most REE pre-diction equations, BMI components such as height and weight were used as variables in the prediction equation, therefore, future studies are needed to determine the accuracy of the REE prediction equation for each BMI group. (Page 8, Lines 306–310)

Comment #3: Although authors compared the error of five equations, however, only Harris-Benedict equation was discussed. Please discuss more why there are different results calculated by different prediction equations

Response: We agree with the reviewer’s comment. Accordingly, we have added relevant sentences to the Discussion section as follows:

The Harris-Benedict equation is commonly used for REE prediction in clinical practice in Japan. However, our study revealed that this equation predicted a value that was, on average, 69.5 kcal/day less than the measured REE. Furthermore, compared to the measured REE, the FAO/WHO/UNU equation and Schofield equation overestimated by 46.2 kcal/day and 97.9 kcal/day, respectively, while the Ganpule equation and body weight × 20 equation underestimated by 138.0 kcal/day and 110.4 kcal/day, respectively. A previous study, that measured REE by indirect calorimetry in 68 hospitalized patients aged 74.4 ± 9.3 years, reported that the predicted REE computed by the Harris-Benedict equation was 26.7 kcal/day lower than the measured REE, displaying a smaller error than our findings [30]. Furthermore, Siervo et al. study, the FAO/WHO/UNU equation was 105 kcal/day higher than the measured REE, while the Schofield equation was 7.2 kcal/day lower, and the accuracy of the Schofield equation seems to be higher. In our findings, both prediction equations were overestimated, and thus, our study showed a different trend from previous studies. Similarly, Neelemaat et al. assessed indirect calorimetry on 194 hospitalized older patients aged 74.3 ± 9.1 years and found that the predicted REE calculated by the Harris-Benedict equation was 141 kcal/day lower than the measured REE, displaying a larger error than our findings. Furthermore, in the Neelemaat et al. study, the FAO/WHO/UNU equation and the Schofield equation showed smaller values than the measured REE. (Page 6, Lines 229–245)

The authors also concluded that the predicted REE using the Ganpule equation had the highest prediction accuracy compared to the Harris-Benedict, Schofield, and FAO/WHO/UNU equations [28]. This study examined the prediction equations presented in the Japan-DRI (Males; body weight × 21.5, Females; body weight × 20.7), which are similar to the body weight × 20 prediction equation. The results reported values approximately 32 to 80 kcal/day greater than the measured REE. Due to the slightly larger coefficients and the different patient settings, the trends were different from our findings. (Page 7, Lines 256–261)

In our study, the Harris-Benedict and FAO/WHO/UNU equations showed relatively high estimation accuracy among the prediction equations analyzed, however, the observed errors were too large for the equations to be used in clinical practice. (Page 7, Lines 262–265)

Therefore, the feasibility of the Harris-Benedict, FAO/WHO/UNU, Ganpule, Schofield, and body weight × 20 as predictive equations for older hospitalized patients seems to be limited, as there are conflicting reports in the literature. (Page 7, Lines 272–274)

Reviewer 2 Report

Recession of manuscript No. 2052914: „ Resting Energy Expenditure in Older Inpatients: A Comparison of Prediction Equations and written by Fumiya Kawase, Yoshiyuki Masaki, Hiroko Ozawa, Manami Imanaka, Aoi Sugiyama, Hironari Wada, Ryokichi Goto, Shinya Kobayashi and Takayoshi Tsukahara, which will be published in Nutrients.

            The structure of manuscript has the commonly required criteria. The topic of presented work is very actual. Determining energy intake is an important component of nutritional support for patients with malnutrition; however, the validity of prediction equations for resting energy expenditure is discussed in older hospitalized patients. Malnutrition is associated with adverse clinical outcomes including prolonged hospital stays, increased in-hospital mortality, and increased in-hospital falls. Thus, interventions for malnutrition have the potential to improve patient care and quality of life and reduce healthcare costs. 

            In the present study, authors aimed to assess the validity of these equations in older hospitalized patients in Japan. This study had a few limitations. First, this was a single-center cross-sectional study conducted in Japan, which limits the generalization of results due to possible selection bias of participants. Second, since the study included individuals with diverse diseases and backgrounds that were not controlled for, it is possible that our results were confounded by such factors. The impact of disease on REE is unknown even for diseases that have a significant impact on metabolism. The results of this study showed that none of the five equations compared had predictive equations with high estimation accuracy in older Japanese hospitalized patients.

The results are documented in table that present the review of the obtained data.  

            The citations are well-chosen and relevant and their format respects usual standards. The conclusion summarizes the author´s results.

Author Response

We appreciate your favorable comments.
